# Quantitative Load Dependency Analysis of Local Trabecular Bone Microstructure to Understand the Spatial Characteristics in the Synthetic Proximal Femur

**DOI:** 10.3390/biology12020170

**Published:** 2023-01-20

**Authors:** Jisun Kim, Bong Ju Chun, Jung Jin Kim

**Affiliations:** 1Department of Mechanical Engineering, Keimyung University, Daegu 42601, Republic of Korea; 2Cho Chun Shik Graduate School of Mobility, Korea Advanced Institute of Science and Technology (KAIST), Daejeon 34051, Republic of Korea

**Keywords:** load dependency analysis, quantitative analysis, strain energy density, proximal femur, region of interest, bone remodeling

## Abstract

**Simple Summary:**

This research aims to quantitatively analyze the trabecular bone microstructure in the synthetic proximal femur based on the region of interest (ROI) to better understand the spatial characteristics under various loading conditions. For this purpose, the load dependencies of twelve ROIs were evaluated under seven different loading conditions. According to the analysis, bone microstructures in the metaphysis and global model depended on multiple loads, whereas those in the epiphysis depended on single or double loads. These findings showed that a specific ROI primarily depends on specific loading conditions. This study provides the foundation for future local bone reconstruction utilizing only specific loads with high load dependency as well as a basis for further studies to validate the actual bone structure.

**Abstract:**

Analysis of the dependency of the trabecular structure on loading conditions is essential for understanding and predicting bone structure formation. Although previous studies have investigated the relationship between loads and structural adaptations, there is a need for an in-depth analysis of this relationship based on the bone region and load specifics. In this study, the load dependency of the trabecular bone microstructure for twelve regions of interest (ROIs) in the synthetic proximal femur was quantitatively analyzed to understand the spatial characteristics under seven different loading conditions. To investigate the load dependency, a quantitative measure, called the load dependency score (LDS), was established based on the statistics of the strain energy density (SED) distribution. The results showed that for the global model and epiphysis ROIs, bone microstructures relied on the multiple-loading condition, whereas the structures in the metaphysis depended on single or double loads. These results demonstrate that a given ROI is predominantly dependent on a particular loading condition. The results confirm that the dependency analysis of the load effects for ROIs should be performed both qualitatively and quantitatively.

## 1. Introduction

For several years, significant effort has been made to understand the formation of the bone structure in the human body for diagnosing and predicting the prognosis of bone disorders such as osteoporosis and osteoporotic fractures [1]. As a preliminary investigation of the internal bone architecture, Ward [2] compared the distinctive trabecular pattern of the human femoral neck to the support bracket of a street lamp. In 1867, Von Meyer [3] and Culmann [4] developed a schematic of the trabecular bone structure of the human proximal femur. Based on this study, Wolff reported that the external stresses on the bone are altered by trauma or changes in life patterns, and the functional adaptation of bones causes the trabeculae to align with the new principal stress trajectories [5,6,7]. In addition, he stated that the bone achieves “maximum” mechanical efficiency with “minimal” mass. One of the most important contributions to musculoskeletal research to date is the conceptual model of the mechanostat, which was proposed by Frost in 1987 [8]. According to this theory, the bone, along with other musculoskeletal tissues such as cartilage, tendons, and muscles, responds to routine exercise and loading. Furthermore, changes in the loading environment lead to an adequate structural adaptation of the bone architecture. In other words, a physiological feedback mechanism in the bones allows them to modify their structure and mass in response to external loads. The change in the distribution of the material characteristics is considered throughout the process of changing the bone architecture, which is called “bone remodeling” in the literature [9].

These studies have become an important basis for analyzing bone formation mechanisms and predicting changes in bone structure [10,11,12,13,14]. Numerous methods based on Wolff’s law have been proposed to predict bone adaptation by simulating the bone metabolism of the human body (i.e., bone remodeling), such as the strain energy density (SED)-based bone remodeling method [15,16,17,18], uniform-stress-based trabecular remodeling method [19,20,21,22], microcrack-healing-based method [23,24,25], and the topology optimization-based method [26,27,28,29,30,31,32,33,34]. Studies involving these methods show that human bones are affected by various loading conditions daily [5,35,36,37]. Therefore, having accurate information on the loading conditions is an important prerequisite for understanding and predicting bone structure formation [38].

The inverse bone remodeling approach, which estimates the loading conditions by inversely applying the principles of bone formation to the given bone morphology, is considered a potential candidate for estimating the loading conditions [9]. One can also estimate the loading history from the given morphology if it is possible to predict the resulting bone morphology from the given loading conditions. Accordingly, Fischer et al. [39] developed an optimization-based load estimation method to determine the most uniform stress distribution and achieve the required bone density for the continuum finite element (FE) model. This method was later used to investigate the proximal femurs of humans [40,41,42], chimpanzees, gorillas, and grizzly bears [43]. Christen et al. [44,45] proposed an algorithm for estimating loading conditions by determining the loading history that results in the most uniform SED distributions on the bone structure. This method was further applied to the subject-specific load estimation for the human distal radius [46]. In recent years, studies have used artificial neural networks [14] to estimate unknown musculoskeletal loads from the known trabecular bone structure. These studies successfully predicted the loading conditions; however, they assumed that the adapted trabecular structure displays globally uniform stress throughout the range of the applied loads.

Conversely, other studies have shown differences in the load adaptability of the bone structure according to the internal location of the bone. Warden et al. [47] explored the spatial adaptation of the proximal femur to physical activity and demonstrated that its adaptability to the same load varies depending on the site. The inferior femoral neck region has high adaptability to an applied load, whereas the superior femoral neck region lacks such adaptability. This is because the trabecular architecture has a complex porosity structure, with site-specific trabecula patterns, needing to be appropriately aligned along the load path. The specific region in the femur exhibits characteristic patterns [48,49,50,51,52] such as principal tensile, principal compressive, greater trochanter, secondary tensile, and secondary compressive groups, which indicate that although many different loads may have been applied to form the global bone microstructure, only a few specific loads may have affected a particular local area. However, studies on the proximal femur often acquire information limited to the specific size of the region of interest (ROI) owing to the limitations in the resolution of imaging devices and actual bone sampling [53,54]. Therefore, to estimate bone loads accurately, the dependency of the trabecular structure on the loading conditions should be analyzed based on the region and the load. The use of multiple loading conditions renders the analysis between the bone microstructure and the loading conditions computationally burdensome [29]. However, among various external loading conditions applied to form the bone, identifying the condition that has a greater effect on a specific region can contribute to computationally efficient bone microstructure analyses by reducing the number of loading conditions applied in a local area.

In addition to the need for methodologies to analyze local load dependency, it is essential to analyze the load dependency quantitatively to effectively assess the dependency of applied loading conditions on the bone. This quantitative knowledge about the stress or strain distribution is crucial for achieving a deeper understanding of the mechanisms of failure relating to osteoporosis, osteoarthritis, and the loosening of implants. SED or stress distribution uniformity is widely used to analyze the adaptability of bones to loading conditions [55,56]. However, the uniformity of the SED distribution has been qualitatively assessed through visual methods. For example, in a study of bone reconstruction in the proximal femur using topology optimization, trabecular structures generated under different loading conditions were compared in a qualitative manner [29]; the SED distribution of the trabecular bone was observed to be more uniform under the three complex loads of one-legged stance, abduction, and adduction compared to the that in single-loading conditions. Moreover, in a study quantifying tissue stresses and strain in the trabeculae of a canine proximal femur [57], the preferred load direction was quantitatively explored by determining a uniform strain distribution using contour plots of the standard deviation of principal strain within the femoral head tissue. However, the analysis was limited to three simple loading conditions and one specific volume of interest. Based on this context, this study hypothesized that a quantitative load dependency analysis from a local perspective on the loading conditions applied to construct the global model would show that the local bone microstructure tends to depend on a specific load rather than the applied entire load. Therefore, a quantitative evaluation that includes more areas of interest and a variety of loading conditions is needed to improve the understanding of bone adaptability to various loading conditions.

Therefore, the objective of this study is to conduct a quantitative load dependency analysis of the trabecular bone microstructure in the proximal femur based on the ROI to improve the understanding of spatial characteristics under various loading conditions. Topology optimization is performed using a two-dimensional (2D) synthesized FE model of the proximal femur with commonly used loading conditions [58,59]. Then, seven combinations of loading conditions are defined and applied to the trabecular structure to analyze its dependency on the applied loading conditions. The load dependency is quantitatively evaluated for each of the twelve ROIs in terms of the load dependency score calculated from the statistical parameters of the SED distribution. Based on the quantitative load dependency analysis, the spatial characteristics of each ROI are further investigated, and the dominant loading condition for each ROI is identified.

## 2. Materials and Methods

The dependency of the trabecular bone microstructure in the synthetic proximal femur on various external loading conditions was quantitatively analyzed based on the ROI, and the dominant loading condition for each ROI was determined in three steps (Figure 1). First, the trabecular microarchitecture of the synthetic proximal femur was generated by topology optimization (Section 2.1). Second, twelve ROIs were selected, and seven loading conditions were generated (Section 2.2). Finally, the load dependency was evaluated based on the ROI against the seven loading conditions (Section 2.3).

### 2.1. Generation of Trabecular Structure for the Synthetic Proximal Femur

A 2D synthetic proximal femur FE model with a resolution of 50 μm [29] (Figure 2a) was selected for the load dependency analysis. The specific bone microstructure and subject-specific loading conditions that are used to form its bone structure are essential to achieve the accurate quantitative load dependency. However, estimating the actual loading condition applied to the real bone structure is still challenging. Therefore, it is difficult to obtain an actual accurate bone-load condition pair. Therefore, a synthetic proximal femur was used to analyze the effect of the known bone microstructure on the accurate known loading conditions.

The synthetic femur model used in this study exhibits structural feasibility in its bone microstructures according to the previous study [29]. Accordingly, the proximal femur has been extensively used as a valid model in numerous investigations such as the application of design space optimization to bone remodeling [60], the study of trabecular changes with aging [32], research on image resolution enhancement [61], an interaction study of solitary waves with the bone [62], and the study of the structural behavior of scaffolds inserted bones [63].

The topology optimization method described in the literature [29] was used to construct the trabecular structure for the load dependency analysis. An initial femur FE model comprising 2,149,488 elements with a resolution of 50 μm was described in the 94.2 mm × 104.4 mm image. A resolution of 50 μm is enough to represent the trabecular bone microstructure with a thickness ranging from 100–200 μm [64]. Moreover, it has been shown that human cancellous bone tissues that were studied using FE analysis converged with a very slight difference of 1.65% at resolutions below 156 μm [65]. Therefore, an element size of 50 μm, which is smaller than the size validated in previous studies [65,66], was used in the present study. The value of Young’s modulus for the bone was assigned based on the solid-isotropic-material-with-penalization method [67], given as follows:(1)$$E_{i}=E_{0}\cdot \rho_{i}^{3},$$
where *E*_0_ = 15 GPa (for trabecular bone) or 22.5 GPa (for cortical bone) [64,65], *ρ_i_* is the bone mass density of the *i*th element, and *E_i_* is the elastic modulus for the *i*th element. The value of Poisson’s ratio for both cortical and trabecular bones is 0.3 [65,66]. The entire lower distal part of the femur model was fixed, and three different load cases were applied to the femoral head and greater trochanter (Figure 2a) for topology optimization. For each loading condition, the hip contact force and muscle force acted on the femoral head and greater trochanter in the form of a distributed load. The three loading conditions [68] were as follows: one-legged stance (6000 cycles per day), abduction (2000 cycles per day), and adduction (2000 cycles per day). These loads were presented by Beaupre et al. [58,59] to represent daily activities [19,21,29,61] (Figure 2b).

The topology optimization-based bone remodeling simulation [29] generated the trabecular structure. The objective function minimized the compliance (i.e., total strain energy stored in the trabecular structure) *f*(**ρ**) considering the constraints of mass and perimeter. The mathematical formulation of the topology optimization is given as follows:(2)$$\begin{array}{ll} \mathrm{Minimize} & f(\rho )=\sum_{j=1}^{M}c_{j}\left(\frac{1}{2}u_{j}{}^{T}Ku_{j}\right)\\ \mathrm{Subject}\;\mathrm{to} & g_{1}(\rho )=\sum_{i=1}^{N}\rho_{i}v_{i}\leq M_{0}\\ & g_{2}(\rho )=P\geq P_{0}\\ & 0.055\leq \rho_{i}\leq 1.0 \end{array},$$
where $\rho ={\left[\rho_{1}\;\rho_{2}\;\cdots \;\rho_{N}\right]}^{T}$ is a design vector for bone density, *c_j_* is a normalized weighting factor for the *j*th load case, **u***_j_* is the nodal displacement vector for the *j*th load case, *M* is the total number of load cases, **K** is a global stiffness matrix, *v_i_* is the volume of the *i*th finite element, *N* is the total number of FEs, and *P* is the perimeter of the bone structure. *M*_0_ and *P*_0_ denote the initial design values for bone mass and perimeter, respectively. All FE analyses were conducted on a personal computer (Intel(R) Core ™ i9-10900K, 3.70 GHz, 128 GB RAM). The FE equation solver in Ansys 2022 R1 (Ansys, Inc., Canonsburg, PA, USA) was used with the preconditioned conjugate gradient method [69], and the optimization was performed using the method of moving asymptotes [70]. Further details can be found in [29].

### 2.2. Selection of Regions of Interest and Loading Conditions

Assuming that a specific location of the trabecular structure primarily depends on a particular load case, twelve ROIs were selected to identify the dominant load case for each ROI location. The femur mainly comprises a cortical bone and a trabecular bone (Figure 3a). The cortical bone surrounds the outer part of the femur; although the thickness is different for each part, it has low porosity and high rigidity overall. The trabecular bone is distributed throughout the proximal femur, and the distribution and direction of porosity varies according to the stiffness of each position. The proximal femur is largely divided into two areas: epiphysis, a region where the load is directly applied, and metaphysis, which has a characteristic that is primarily affected by the load transmitted by the trabecular bone. Therefore, for ROI selection, both the epiphysis and metaphysis regions with different characteristics were considered in this study.

Twelve ROIs (Figure 3b) based on five different trabecular groups demonstrated by Singh et al. [48,50] were selected to best express the main trabecular pattern of the proximal femur (Figure 3a). ROIs 1, 2, and 3 are the main locations where adduction, one-legged stance, and abduction loading conditions act on the femoral head, respectively. ROIs 4, 5, and 6 are locations that represent the femoral neck: ROI 4 is the location close to the femoral head where the principal tensile group and principal compressive group intersect; ROI 5 is the inferior femoral neck region where the trabecular bone in the principal compressive direction is thickest and most distinct; and ROI 6 is in Ward’s triangle [2], which has a low trabecular bone density in the femoral neck. ROIs 7 and 8 are the greater trochanter regions that best represent the greater trochanter group; ROIs 7 and 8 near the greater trochanter and the cortical bone, respectively. ROIs 9, 10, 11, and 12 are located at the top, right, left, and bottom of the intertrochanteric region, respectively. ROIs 9 and 12 are located at the intersections of the secondary compressive and tensile groups in the middle of the intertrochanteric region. ROIs 10 and 11 are located near the cortical bone, where the secondary compressive and tensile groups begin to expand. These ROIs are classified as belonging to either the epiphysis or the metaphysis of the proximal femur structure (Figure 3a). The epiphysis includes ROIs 1–4, 7, and 8, whereas the metaphysis includes ROIs 5, 6, and 9–12.

In this study, the ROI size was selected as 9.6 mm × 9.6 mm to sufficiently express the characteristics of the bone microstructure for each ROI. Several analyses have been conducted for a size of 10 mm ROIs [53,54,61,68]. Chen et al. [54] used an 8 mm specimen and micro-CT images to analyze the structural characteristics of the 3D bone microstructure. Cui et al. [53] used an 8 mm specimen and a height of 10 mm micro-CT image to analyze the age-dependent changes in the 3D bone microstructure. Rietbergen et al. [71] used a 7 mm 3D FE model to determine trabecular bone elastic properties and loads. A previous study [61] used a 10.2 mm 2D FE model for bone microstructure reconstruction. Therefore, in this study, similar sizes were selected to sufficiently express the characteristics of the bone microstructure for each ROI.

The three load cases [58,59] used for the topology optimization of the bone remodeling model were used as the basis for seven new load cases, which are listed in Table 1. Load cases (load cases) 1–3 are single-loading conditions that correspond to one-legged standing, abduction, and adduction, respectively. Load cases 4–6 are dual-loading conditions comprising two individual loading conditions each, and load case 7 is a multiple-loading condition containing all three of the individual loading conditions. The normalized weights were 1.0 and 0.5 for load cases 1–3 and 0 4–6, respectively. For load case 7, the weights were set differently: *c*_1_ (for the one-legged stance) was 0.6, *c*_2_ (for abduction) was 0.2, and *c*_3_ (for adduction) was 0.2. Load case 7 comprised the same loading conditions used for the topology optimization (described in Section 2.1).

### 2.3. Analysis of Results by ROI for Load Dependency Evaluation

Herein, we investigated the load dependency of the trabecular structure for the ROIs and load cases determined in Section 2.2. Each load case produced a SED distribution in the 2D femur FE model. Then, the statistical parameters of the SED distributions (i.e., average and standard deviation) were used to quantify the effect of each load case condition on the ROIs. The average (mean) of the SED distribution in an ROI expresses how well the loading conditions transferred from the applied location to the ROI, and the standard deviation corresponds to how well the loading conditions adapt to the ROI. Note that the mechanical stimulus (in this study, SED) induces bone remodeling, which in turn eliminates the local stress non-uniformity by changing the trabecular structure.

A quantitative measure called the load dependency score (LDS) was established to investigate the average and standard deviation of the SED simultaneously, calculated for each ROI for the load dependency analysis. The mathematical formulation of LDS is given as follows:(3)$$Avg.S_{i,j}=\frac{\mu_{i,j}-\mu_{i,\mathrm{avg}}}{\sigma_{i,\mathrm{avg}}},$$
(4)$$SD.S_{i,j}=\frac{\sigma_{i,j}-\mu_{i,\mathrm{SD}}}{\sigma_{i,\mathrm{SD}}},$$
(5)$$LDS_{i,j}=100\cdot \left[w\cdot Avg.S_{i,j}-\left(1-w\right)\cdot SD.S_{i,j}\right],$$
where *μ_i,j_* is the average SED in the *i*th ROI under the *j*th loading condition, *σ_i,j_* is the standard deviation of SED in the *i*th ROI under the *j*th loading condition, *μ_i_*_,avg_ is the averaged value of *μ_i,j_* across all *j* in *i*th ROI, *σ_i_*_,avg_ is the averaged value of *σ_i,j_* across all *j* in *i*th ROI, *μ_i_*_,SD_ is the standard deviation of *μ_i,j_* across all values of *j*, *σ_i_*_,SD_ is the standard deviation of *σ_i,j_* across all values of *j*, *Avg. S_i,j_*, and *SD. S_i,j_* are the normalized average and standard deviations, respectively, of the SED in the *i*th ROI under the *j*th loading condition, *LDS_i,j_* is the LDS for the *i*th ROI under the *j*th loading condition, and *w* is a weighting coefficient. The average and standard deviation values were normalized to eliminate the effect of differences in the SED magnitude between ROIs. Consequently, the large magnitude and small standard deviation of the SED generates high LDS values.

The weighting coefficient, *w*, was selected as 0.51 through careful qualitative analysis. As the global femur model was constructed using load case 7, the LDS of the global model under load case 7 should exhibit the highest value among all load cases. This result has been validated in previous studies [29,61,62]. Additionally, among various dual-load conditions, load case 4 showed the most similar SED distribution to load case 7 and exhibited the highest load dependency of bone microstructure. Therefore, a weighting coefficient of 0.51 was selected so that the LDS under load case 4 exhibited the highest value among the dual-load conditions and a lower LDS value than load case 7 simultaneously. This weighting coefficient grants nearly identical importance to the average and standard deviation of the SED in the ROI, the average being given slightly greater importance.

## 3. Results

The statistical values of the SED distributions and the scores for the global model are presented in Table 2. Figure 4 shows the comparison of LDSs for the ROIs in the metaphysis and epiphysis under each load case. The statistical values of the SED distributions and the scores for the ROIs in the epiphysis and metaphysis are presented in Table 3 and Table 4, respectively. The SED distribution of the global model and the score calculation includes all areas of the model, and the calculation of the ROI model includes the applied area of the model. As shown in Table 2, the highest LDS for the global model was found in load case 7 (*LDS*: 68.88). In other words, the bone microstructure in the global model strongly depended on the multiple-loading condition (i.e., load case 7). As shown in Table 3 and Figure 4a, the LDSs for most ROIs located in the epiphysis were highest under load case 7, similar to the global model, thereby indicating that the bone microstructure of the epiphysis strongly depends on the multiple-loading condition. Furthermore, the LDSs for ROIs 7 and 8 were very high in load cases 4 (*LDS*: 55.54) and 6 (*LDS*: 11.21), respectively. Moreover, their error rates against the values of load case 7 were only 1.77% and 2.56%, respectively. In contrast, the highest LDSs for ROIs located in the metaphysis were induced under other load cases. Their dependency was highest on a single or double load rather than the multiple loads of load case 7 (Table 4 and Figure 4b). For ROIs 5, 6, and 11, the highest dependency was found in dual-loading conditions: load case 4 (*LDS*: 15.51), load case 4 (*LDS*: 34.67), and load case 6 (*LDS*: 11.38), respectively. For ROIs 10 and 12, the highest dependency was on the single-loading condition load case 2 (*LDS*: 59.82 and *LDS*: 47.17, respectively), which indicated the existence of a dominant single load for ROIs 10 and 12.

The load dependency of the bone microstructure for different ROIs under different loading conditions can be compared quantitatively using the scores defined in this study (i.e., *Avg. S_i,j_*, *SD. S_i,j_*, and *LDS*). *LDS* and *SD. S_i,j_* provided quantitative results that allowed the unambiguous distinction of the load dependency, which is difficult to evaluate using qualitative assessments (e.g., the evaluation of uniformity by eye alone). Figure 5, Figure 6 and Figure 7 show the SED distribution contours for the global model, ROIs in the epiphysis, and the ROIs in the metaphysis, respectively, which help in the qualitative analysis. The quantitative analysis of the global model yielded the highest *LDS* (68.88) and lowest standard deviation score (−1.16) under load case 7 (Table 2), thereby indicating that the load dependency is relatively high under load case 7 compared to that under other loading conditions and that the SED distribution is uniform with respect to this. In contrast, it is difficult to determine the load dependency using a qualitative assessment wherein loading condition among load cases 1, 4, and 7 induces the dominant and most uniform SED distribution (Figure 5). Moreover, quantitative measurements allow the effects of load to be identified for epiphysis and metaphysis ROIs. As a representative example, the SED distributions of load cases 4 and 7 in ROI 5 (metaphysis) are quite similar, making it nearly impossible to visually distinguish which one induces the greater load dependency (Figure 6). However, the quantitative analysis clearly shows that ROI 5 has relatively high and low load dependencies under load cases 4 (with an LDS of 15.51) and 7 (with an LDS of 10.02), respectively (Table 4). Additionally, the SED distribution is relatively uniform under load case 4 (with an *SD. S*_5,4_ value of 0.28) and is quantitatively expressed as being relatively non-uniform under load case 7 (with an *SD. S*_5,7_ value of 0.38).

The effect of external loads on the ROI transferred by the bone microstructure was clearly demonstrated by the *LDS*, which considers the SED average as well as its standard deviation. A low LDS value was evident when the load effect was small because the external load was not transferred to the ROI. The low influence of the load under specific loads for each ROI can be verified visually using qualitative measurements. As a representative example, the overall average of the SED distribution in ROI 1 under load case 2 was relatively small, as shown in Figure 7. As shown in Table 3, this ROI exhibited a very low LDS value (−35.13), which effectively and quantitatively expressed that the load dependency was very low. Moreover, the low *SD. S*_1,2_ value (−1.18) and low *Avg. S*_1,2_ value (−1.83) indicated that the SED distribution was rather uniform, however, the final load dependency was low owing to the insignificant influence of the load transferred to the ROI. Conversely, when the load was effectively transmitted to the ROI and the SED was uniformly distributed, as in ROI 2 under load case 7 (Figure 7), the *LDS* was relatively high (here, 77.01, as shown in Table 3). This is because *SD. S*_2,7_ has a very low value (−0.95) and *Avg. S*_2,7_ has a very high value (0.60) (Table 3) simultaneously.

## 4. Discussion

The uniformity of stress or SED distribution for the global model (i.e., the full femur model) has been previously investigated to determine the loading conditions and understand the load dependency. However, it is necessary to investigate the uniformity according to the load and region to obtain insights regarding the relationship between the trabecular structure and the loading conditions. Therefore, this study analyzed the trabecular structures for twelve ROIs in the synthetic proximal femur under seven loading conditions to determine the dominant loading condition of the trabecular architecture. The adaptability of the trabecular bone to the conditions of the seven load cases was analyzed using LDS, and the results showed that the trabecular bone in specific ROIs exhibited adaptability to single-loading conditions or dual-loading conditions although the structure was constructed under a multiple-loading condition. This single- or double-load adaptability was more prominent in the metaphysis ROIs compared to the epiphysis ROIs.

This study suggested that the average and standard deviation of the SED distribution are required to precisely evaluate the bone’s adaptability to the loading conditions in the ROI. To date, many studies have utilized stress uniformity or the uniformity of the SED distribution as the standard for evaluating the adaptability of the trabecular bone to loading conditions. For example, the non-uniformity of a SED distribution is considered a reason for reconstructing the trabecular bone structure in various bone remodeling simulation methods [15,19,21,29], and a highly uniform SED distribution indicates that the bone was well adapted to the external load. However, this study demonstrated that if the average of the SED distribution is low, the trabecular bone structure in an ROI cannot be considered adapted to a given loading condition even if the SED is uniformly distributed in the ROI. For example, in ROI 1 in Table 3 and Figure 4a, the standard deviation of a SED distribution (1.23 × 10^−2^ J/mm^2^) was the lowest in load case 2. If the load dependency is analyzed using the uniformity of SED alone as the conventional analysis, the bone microstructure of ROI 1 will be most dependent on load case 2. However, as shown in Figure 5 and ROI 1 under loading condition 2 in Figure 7, the SED delivery path of load case 2 almost did not pass through the location of ROI 1, which indicates that load case 2 did not impart any internal stress to the specific location, ROI 1, and hence, had little or no effect on the bone microstructure formation in ROI 1. This phenomenon is expressed as the average of the SED distribution (7.82 × 10^−3^ J/mm^2^) of ROI 1 in Table 4, and Figure 4b shows the lowest value in load case 2 among load cases. Therefore, to accurately evaluate the adaptability of the bone microstructure to the applied loading condition in the ROI, the following factors must be considered together: (1) the average of the SED distribution, which indicates whether the load is sufficiently well delivered from the external load to the ROI, and (2) the standard deviation of the SED distribution, which expresses the degree of adaptability to the load delivered. These findings indicate that LDS, the adopted metric proposed in this study, can help understand the mechanical relationship between the applied load and the bone microstructure in the ROI. This suggests that the sole use of the uniformity of the SED distribution to assess the load dependency of the structure can lead to misdiagnosis.

Furthermore, this study proposed that the SED distributions should be compared qualitatively as well as quantitatively to accurately assess the uniformity of the SED distribution for the trabecular bone under given loading conditions. Conventionally, bone remodeling simulation methods have evaluated the uniformity of SED distributions qualitatively to analyze the adaptability of the trabecular bone structure to the applied loading conditions [15,19,21,29]. Indeed, it can be roughly observed that the SED is almost uniformly distributed over the trabecular structure (Figure 5, Figure 6 and Figure 7). However, the qualitative result does not show the load case that significantly affects the trabecular structure. For example, as shown in Figure 7, ROI 2 had a uniform SED distribution under both load cases 6 and 7, however, it is difficult to identify which has the greater uniformity. Similarly, ROI 3 had a uniform SED distribution under both load cases 6 and 7, however, it is difficult to compare their relative uniformities. Conversely, the quantitative analysis clearly identified the SED distribution that was more uniform under a specific loading condition through the standard deviation (Table 3). For ROI 2, the standard deviations of the SED distribution under load cases 6 and 7 were 3.63 × 10^−2^ J/mm^2^ and 3.61 × 10^−2^ J/mm^2^, respectively, which are nearly identical values. For ROI 3, the standard deviations of the SED distribution under load cases 6 and 7 were 3.99 × 10^−2^ J/mm^2^ and 3.52 × 10^−2^ J/mm^2^, respectively, indicating that the SED distribution under load case 7 was more uniform than that under load case 6.

Based on the load dependency analysis, it can be assured that bone remodeling occurring in a specific region of the femur depends highly on a particular loading condition, called the dominant load in this study. The synthetic proximal femur model used for the analysis was created using a topology optimization bone remodeling simulation method by applying a multiple-loading condition that represents daily activities. Therefore, the LDS for the global model was the highest under load case 7 (Table 2), indicating that the global model was best adapted to the multiple-loading condition. This result agrees well with the assumption from the inverse method that the adapted trabecular structure displays a globally uniform stress throughout the range of the applied loads [44,45,46]. However, although the global proximal femur model was adopted under load case 7 (i.e., the multiple-loading condition used for topology optimization), several ROIs were adapted to other load cases, such as ROI 10 and 12, which were adapted to load case 2. In particular, load cases 4 and 2 had particularly large influences on the femoral neck (ROI 5) and the intertrochanteric region (ROIs 10 and 12), respectively. Adachi et al. presented the results of single-load and multiple-load bone remodeling models as figures [19]. From the results of the present study (Figure 8), it can be confirmed that the trabecular structure on the right side of the intertrochanteric region adapted to abduction and that the trabecular structure adapted to the multiple loads was similar. The results from the two studies are mutually consistent and indicate that bone remodeling is dependent on the specific load at a specific region.

This study demonstrated that a given ROI depends on a particular loading condition, which indicates that the localized bone structure can be generated by the dominant single (or dual) loading condition. This fact could be used to improve the simulation of computational bone remodeling (or bone microstructure reconstruction) based on topology optimization. For example, knowledge of the single dominant loading condition for each ROI can help improve the computational efficiency of topology-optimization-based bone remodeling simulation, which consumes considerable computational resources and time because of the large number of finite elements and analysis iterations required. If the dominant load is determined and only the relevant ROI is analyzed, accurate results can be obtained using fewer computational resources and lesser time compared to the case when analyzing the global model under conditions of multiple loads. In clinical practice, information of a specific area sensitive to skeletal disease (i.e., the femoral neck) is typically used rather than that of the entire femur area [69,70]; hence, the proposed method is likely to be suitable for clinical application. Therefore, the expenditure of computational resources and time for bone remodeling simulation for an ROI can supposedly be improved if the analysis can be carried out using a reduced number of loading conditions.

This study has some limitations. First, this study did not investigate all combinations of loading conditions but rather compared the load dependency for only a few hypothesized specific load cases. However, using the hypothesized and limited load cases, it was found that although the global model was constructed using three loading conditions, a dominant load from among these affects a given ROI. Therefore, the set of load cases used in this study is meaningful because it was sufficient to suggest the necessity of conducting the load dependency analysis by ROI. Second, the value of LDS, the principal quantitative score used in this study, is not an absolute measure of load dependency. For example, the fact that the *LDS* of 77.01 for ROI 2 under load case 7 is greater than the *LDS* of 10.93 for ROI 8 under load case 7 (Table 3 and Figure 4a) does not indicate that ROI 2 has a greater load dependency than ROI 8. This is because the LDS is calculated from the standard normal distribution of the results for the load cases in the same ROI. However, it is meaningful as a quantitative value to compare various load cases within an ROI. Third, the FE analysis in this study was based on a 2D model, whereas the real femur is 3D. The final shape of the trabecular bone in the topology optimization matches the shape of the actual trabecular bone even in the 2D model, as shown in a study by Jang et al. [29]. Additionally, it was found that the 2D model is more sensitive to reduced thickness and number of trabecular bones compared to the 3D model. Therefore, analysis using a 2D model can help analyze the relationship between loading conditions and bone microstructure. Finally, this study did not determine whether the dominant loading condition constructs trabecular bone in the same shape in the ROI as in the global model. However, as a proof of concept, this study aims to quantitatively analyze the load dependency of bone microstructure in a specific ROI for the various loading conditions. Therefore, further studies should be conducted to determine the dominant load conditions in the region of interest through the load dependency analysis proposed in this study, and bone microstructure reconstruction should be performed for the ROI to which it is applied. Then, the dominant load can be validated based on whether the ROI obtains the same shape as the global model.

## 5. Conclusions

This study quantitatively analyzed the uniformity of the SED distribution for each ROI in the synthetic femur model under various loading conditions. The LDS was used as a measure of the dependency between the trabecular bone structure and a specific loading condition. The results showed that the trabecular structure in a given ROI primarily depended on a particular loading condition, called the dominant loading condition. Furthermore, it was confirmed that the uniformity of the SED distribution and the average SED value should be considered simultaneously when analyzing the load effects both qualitatively and quantitatively. The results of the analyses provided an in-depth understanding of the spatial characteristics of each ROI. Furthermore, these findings can help improve the computational efficiency of bone remodeling simulation methods. As a dominant load is used rather than several loading conditions, this work may increase the computational efficiency of bone remodeling simulation approaches. As a proof of concept, this study provides a basis for local bone reconstruction using only specific loads with high load dependency in the future and a basis for expanding the study to the validation of real bone.

## Figures and Tables

**Figure 1 biology-12-00170-f001:**
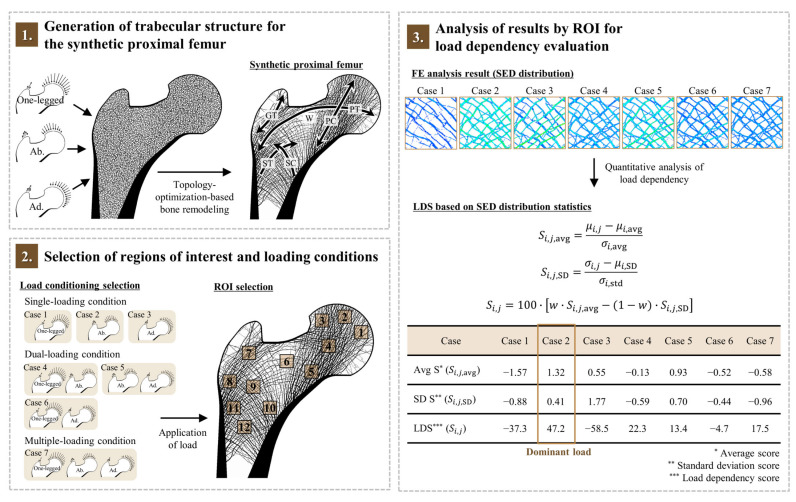
Overview of loading condition dependency analysis of the trabecular bone microstructure based on the ROI for different loading conditions. Ab: abduction; Ad: adduction; GT: the greater trochanter group; PC: the principal compressive group; PT: the principal tensile group; ST: the secondary tensile group; SC: the secondary compressive group; W: Ward’s triangle; ROI: region of interest; FE: finite element; SED: strain energy density; LDS: load dependency score; and S: score.

**Figure 2 biology-12-00170-f002:**
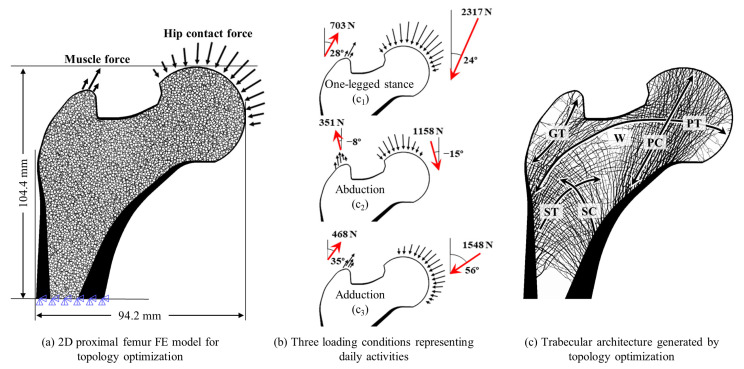
Generation of trabecular bone microstructure based on topology optimization. (**a**) Finite element model with the initial pattern of hollow circles, (**b**) three loading conditions: one-legged stance, abduction, and adduction, and (**c**) trabecular architecture generated by topology optimization.

**Figure 3 biology-12-00170-f003:**
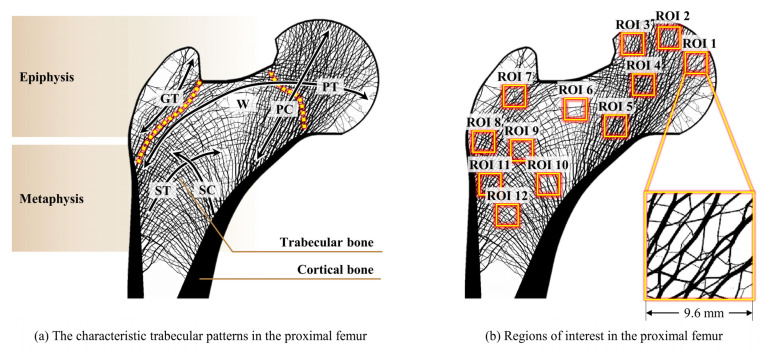
Characteristic trabecular patterns and 12 ROIs in the human proximal femur. (**a**) Important anisotropic loading groups: GT: the greater trochanter group; PC: principal compressive group; PT: principal tensile group; SC: secondary compressive group; ST: secondary tensile group; and W: Ward’s triangle; and (**b**) sizes and locations of the selected ROIs considering the characteristic patterns.

**Figure 4 biology-12-00170-f004:**
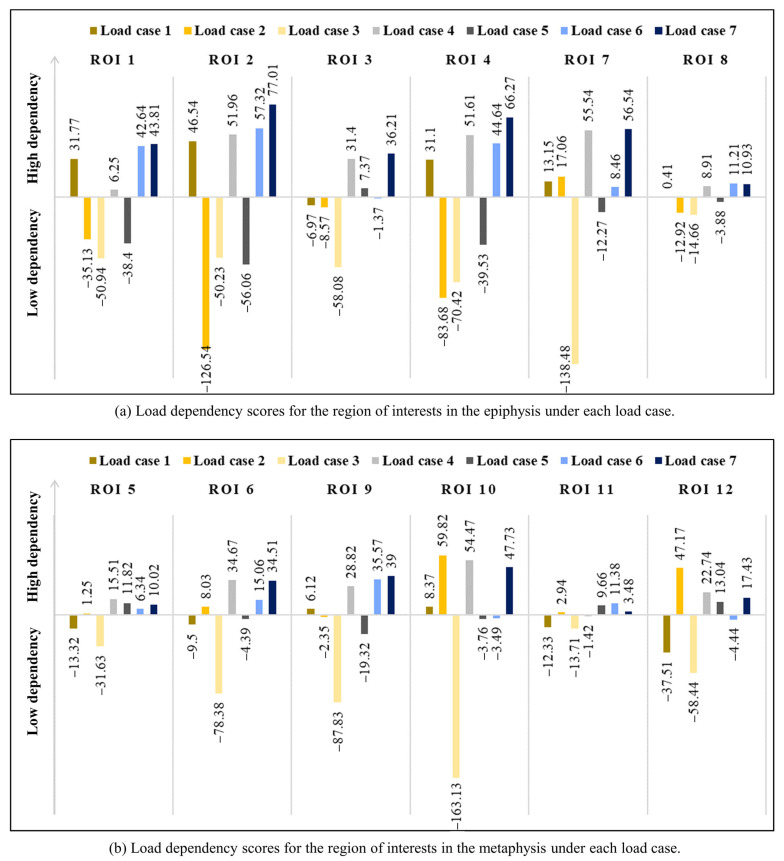
Load dependency scores for the region of interests under each load case. (**a**) Load dependency scores for the region of interests in the epiphysis under each load case, and (**b**) Load dependency scores for the region of interests in the metaphysis under each load case.

**Figure 5 biology-12-00170-f005:**
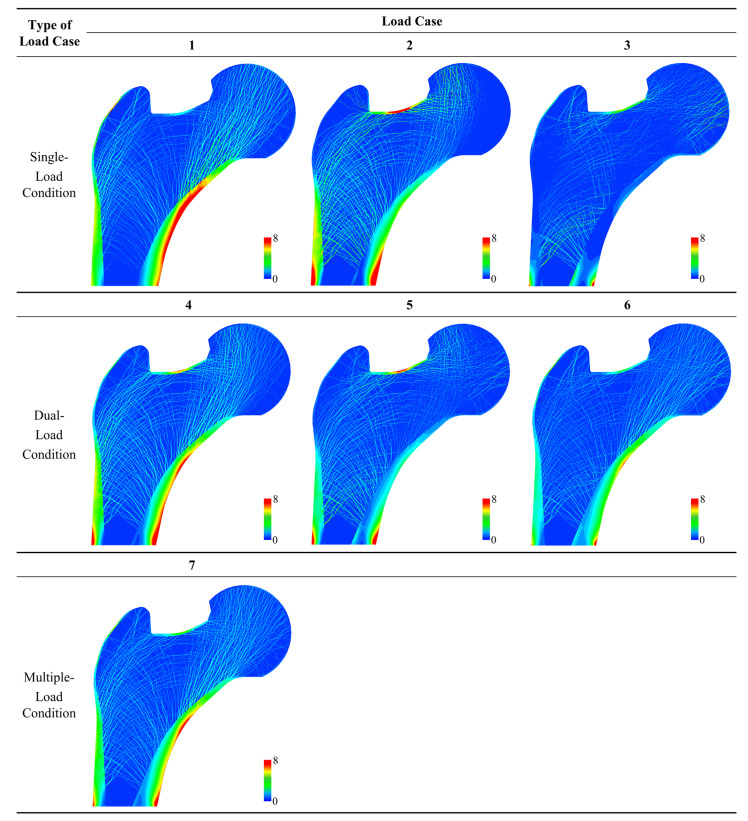
Strain energy density (J/mm^2^) contours for global model for each load case.

**Figure 6 biology-12-00170-f006:**
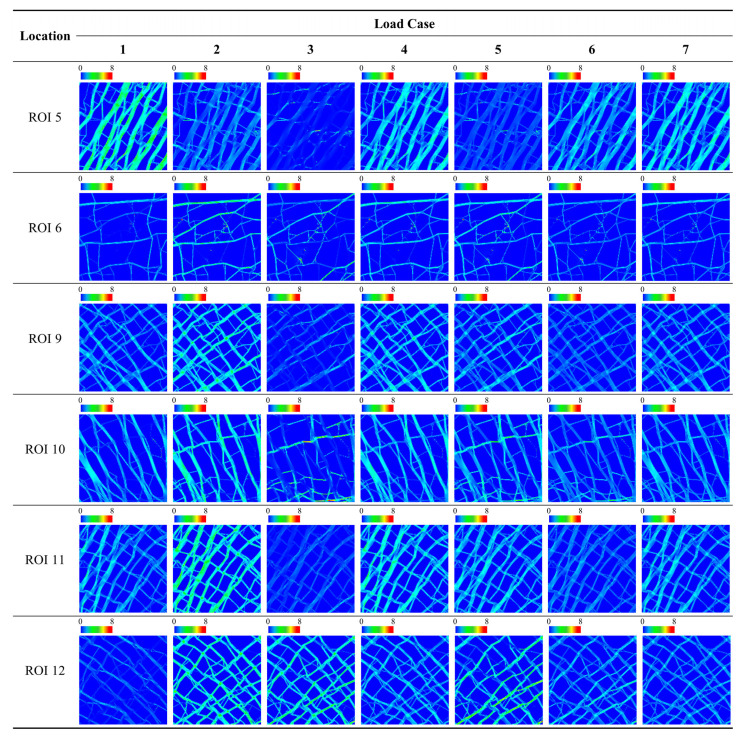
Strain energy density (J/mm^2^) contour plots for ROIs in the metaphysis for each load case.

**Figure 7 biology-12-00170-f007:**
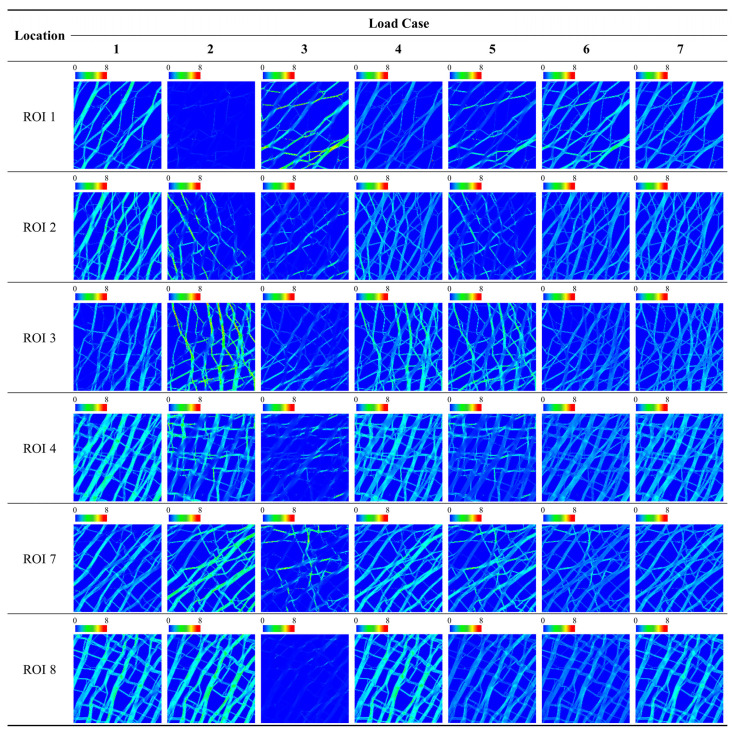
Strain energy density (J/mm^2^) contour plots for ROIs in the epiphysis for each load case.

**Figure 8 biology-12-00170-f008:**
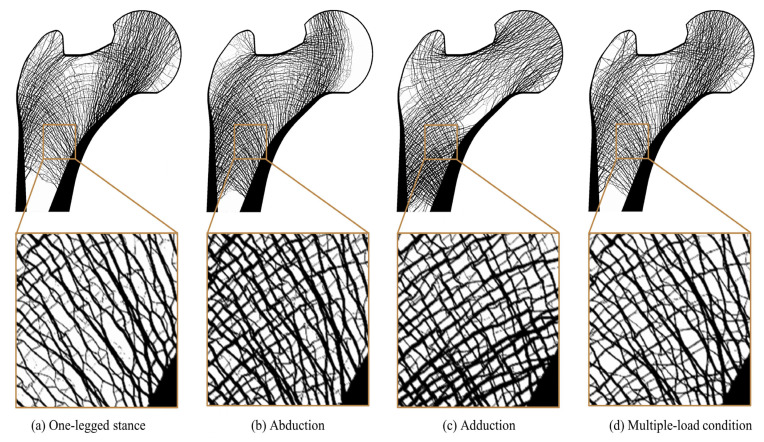
Morphological change in trabecular architecture of proximal femur for (**a**–**c**) single-loading conditions and (**d**) multiple-loading condition.

**Table 1 biology-12-00170-t001:** Loading conditions for proximal femur structural behavior analysis.

Type of Load Case	Load Case	Loading Condition	Normalized Weight
Single-loading condition	1	One-legged stance	*c*_1_ = 1.0
	2	Abduction	*c*_2_ = 1.0
	3	Adduction	*c*_3_ = 1.0
Dual-loading condition	4	One-legged stance	*c*_1_ = 0.5
		Abduction	*c*_2_ = 0.5
	5	Abduction	*c*_2_ = 0.5
		Adduction	*c*_3_ = 0.5
	6	One-legged stance	*c*_1_ = 0.5
		Adduction	*c*_3_ = 0.5
Multiple-loading condition	7	One-legged stance	*c*_1_ = 0.6
		Abduction	*c*_2_ = 0.2
		Adduction	*c*_3_ = 0.2

**Table 2 biology-12-00170-t002:** Results of the statistical analysis for strain energy density and scores for global model for each load case.

Location	Category	Metric	Load Case						
			1	2	3	4	5	6	7
Global	SED	*Avg.* [J/mm^2^]	1.37 × 10^−1^	1.47 × 10^−1^	7.20 × 10^−2^	1.42 × 10^−1^	1.09 × 10^−1^	1.05 × 10^−1^	1.26 × 10^−1^
		*SD* [J/mm^2^]	7.69 × 10^−2^	1.10 × 10^−1^	9.19 × 10^−2^	6.11 × 10^−2^	7.85 × 10^−2^	4.75 × 10^−2^	4.60 × 10^−2^
	Score	*Avg. S* *	0.65	1.03	−1.81	0.84	−0.41	−0.56	0.24
		*SD. S* **	0.16	1.57	0.80	−0.51	0.23	−1.09	−1.16
		*LDS* ***	25.48	−24.45	−131.38	68.18	−31.92	25.20	68.88

* Average score; ** Standard deviation score; *** Load dependency score.

**Table 3 biology-12-00170-t003:** Result of statistical analysis for strain energy density and scores for ROIs in the epiphysis for each load case.

Location	Category	Metric	Load Case						
			1	2	3	4	5	6	7
ROI 1	SED	*Avg.* [J/mm^2^]	1.46 × 10^−1^	7.82 × 10^−3^	1.48 × 10^−1^	7.71 × 10^−2^	7.81 × 10^−2^	1.47 × 10^−1^	1.19 × 10^−1^
		*SD* [J/mm^2^]	7.40 × 10^−2^	1.23 × 10^−2^	1.51 × 10^−1^	3.61 × 10^−2^	7.76 × 10^−2^	6.50 × 10^−2^	3.91 × 10^−2^
	Score	*Avg. S* *	0.82	−1.83	0.85	−0.50	−0.48	0.84	0.30
		*SD. S* **	0.20	−1.18	1.93	−0.65	0.28	0.00	−0.58
		*LDS* ***	31.77	−35.13	−50.94	6.25	−38.40	42.64	43.81
ROI 2	SED	*Avg.* [J/mm^2^]	1.63 × 10^−1^	6.30 × 10^−2^	4.57 × 10^−2^	1.13 × 10^−1^	5.43 × 10^−2^	1.04 × 10^−1^	1.20 × 10^−1^
		*SD* [J/mm^2^]	7.42 × 10^−2^	9.86 × 10^−2^	5.36 × 10^−2^	4.38 × 10^−2^	6.11 × 10^−2^	3.63 × 10^−2^	3.61 × 10^−2^
	Score	*Avg. S* *	1.61	−0.75	−1.16	0.43	−0.95	0.22	0.60
		*SD. S* **	0.73	1.80	−0.18	−0.61	0.15	−0.94	−0.95
		*LDS* ***	46.54	−126.54	−50.23	51.96	−56.06	57.32	77.01
ROI 3	SED	*Avg.* [J/mm^2^]	9.77 × 10^−2^	1.88 × 10^−1^	7.59 × 10^−2^	1.43 × 10^−1^	1.32 × 10^−1^	8.68 × 10^−2^	1.11 × 10^−1^
		*SD* [J/mm^2^]	5.68 × 10^−2^	1.59 × 10^−1^	7.56 × 10^−2^	7.50× 10^−2^	8.30 × 10^−2^	3.99 × 10^−2^	3.52 × 10^−2^
	Score	*Avg. S* *	−0.56	1.78	−1.12	0.62	0.33	−0.84	−0.21
		*SD. S* **	−0.44	2.03	0.02	0.01	0.20	−0.85	−0.96
		*LDS* ***	−6.97	−8.57	−58.08	31.40	7.37	−1.37	36.21
ROI 4	SED	*Avg.* [J/mm^2^]	1.69 × 10^−1^	1.14 × 10^−1^	3.38 × 10^−2^	1.41 × 10^−1^	7.37 × 10^−2^	1.01 × 10^−1^	1.31 × 10^−1^
		*SD* [J/mm^2^]	6.36 × 10^−2^	8.21 × 10^−2^	4.52 × 10^−2^	4.51 × 10^−2^	5.03 × 10^−2^	3.15 × 10^−2^	3.59 × 10^−2^
	Score	*Avg. S* *	1.33	0.11	−1.68	0.71	−0.79	−0.18	0.49
		*SD. S* **	0.75	1.82	−0.31	−0.31	−0.01	−1.10	−0.84
		*LDS* ***	31.10	−83.68	−70.42	51.61	−39.53	44.64	66.27
ROI 7	SED	*Avg.* [J/mm^2^]	1.20 × 10^−1^	1.83 × 10^−1^	6.61 × 10^−2^	1.51 × 10^−1^	1.24 × 10^−1^	9.33 × 10^−2^	1.22 × 10^−1^
		*SD* [J/mm^2^]	5.49 × 10^−2^	9.37 × 10^−2^	9.24 × 10^−2^	5.47 × 10^−2^	6.96 × 10^−2^	3.99 × 10^−2^	3.55 × 10^−2^
	Score	*Avg. S* *	−0.07	1.60	−1.50	0.75	0.03	−0.78	−0.02
		*SD. S* **	−0.34	1.32	1.26	−0.35	0.28	−0.99	−1.18
		*LDS* ***	13.15	17.06	−138.48	55.54	−12.27	8.46	56.54
ROI 8	SED	*Avg.* [J/mm^2^]	1.60 × 10^−1^	1.81 × 10^−1^	1.12 × 10^−2^	1.70 × 10^−1^	9.61 × 10^−2^	8.57 × 10^−2^	1.34 × 10^−1^
		*SD* [J/mm^2^]	5.94 × 10^−2^	7.23 × 10^−2^	1.33 × 10^−2^	5.94 × 10^−2^	3.87 × 10^−2^	2.88 × 10^−2^	4.59 × 10^−2^
	Score	*Avg. S* *	0.67	1.02	−1.81	0.84	−0.39	−0.57	0.24
		*SD. S* **	0.69	1.33	−1.58	0.69	−0.33	−0.82	0.02
		*LDS* ***	0.41	−12.92	−14.66	8.91	−3.88	11.21	10.93

* Average score; ** Standard deviation score; *** Load dependency score.

**Table 4 biology-12-00170-t004:** Result of statistical analysis for strain energy density and scores for ROIs in the metaphysis for each load case.

Location	Category	Metric	Load Case						
			1	2	3	4	5	6	7
ROI 5	SED	*Avg.* [J/mm^2^]	2.16 × 10^−1^	9.41 × 10^−2^	2.75 × 10^−2^	1.55 × 10^−1^	6.08 × 10^−2^	1.22 × 10^−1^	1.54 × 10^−1^
		*SD* [J/mm^2^]	8.55 × 10^−2^	4.00 × 10^−2^	3.17 × 10^−2^	5.40 × 10^−2^	2.49 × 10^−2^	4.70 × 10^−2^	5.59 × 10^−2^
	Score	*Avg. S* *	1.53	−0.38	−1.43	0.57	−0.91	0.06	0.56
		*SD. S* **	1.87	−0.42	−0.84	0.28	−1.18	−0.07	0.38
		*LDS* ***	−13.32	1.25	−31.63	15.51	11.82	6.34	10.02
ROI 6	SED	*Avg.* [J/mm^2^]	1.06 × 10^−1^	1.81 × 10^−1^	9.45 × 10^−2^	1.44 × 10^−1^	1.38 × 10^−1^	1.00 × 10^−1^	1.19 × 10^−1^
		*SD* [J/mm^2^]	5.72× 10^−2^	1.02 × 10^−1^	7.79 × 10^−2^	6.52 × 10^−2^	7.73 × 10^−2^	4.28 × 10^−2^	4.79 × 10^−2^
	Score	*Avg. S* *	−0.66	1.80	−1.03	0.59	0.39	−0.85	−0.23
		*SD. S* **	−0.49	1.71	0.52	−0.10	0.50	−1.19	−0.94
		*LDS* ***	−9.50	8.03	−78.38	34.67	−4.39	15.06	34.51
ROI 9	SED	*Avg.* [J/mm^2^]	1.15 × 10^−1^	1.79 × 10^−1^	5.25 × 10^−2^	1.47 × 10^−1^	1.16 × 10^−1^	8.40 × 10^−2^	1.16 × 10^−1^
		*SD* [J/mm^2^]	4.05 × 10^−2^	5.84 × 10^−2^	4.37 × 10^−2^	4.40 × 10^−2^	4.59 × 10^−2^	2.67 × 10^−2^	3.41 × 10^−2^
	Score	*Avg. S* *	−0.02	1.55	−1.55	0.77	0.01	−0.78	0.01
		*SD. S* **	−0.14	1.66	0.18	0.21	0.40	−1.53	−0.79
		*LDS* ***	6.12	−2.35	−87.83	28.82	−19.32	35.57	39.00
ROI 10	SED	*Avg.* [J/mm^2^]	1.21 × 10^−1^	1.79 × 10^−1^	8.79 × 10^−2^	1.50 × 10^−1^	1.33 × 10^−1^	1.05 × 10^−1^	1.26 × 10^−1^
		*SD* [J/mm^2^]	5.31 × 10^−2^	7.53 × 10^−2^	1.06 × 10^−1^	5.48 × 10^−2^	6.82 × 10^−2^	4.59 × 10^−2^	3.89 × 10^−2^
	Score	*Avg. S* *	−0.26	1.69	−1.38	0.71	0.14	−0.80	−0.10
		*SD. S* **	−0.45	0.53	1.89	−0.37	0.22	−0.76	−1.07
		*LDS* ***	8.37	59.82	−163.13	54.47	−3.76	−3.49	47.73
ROI 11	SED	*Avg.* [J/mm^2^]	1.17 × 10^−1^	2.16 × 10^−1^	5.46 × 10^−2^	1.66 × 10^−1^	1.35 × 10^−1^	8.57 × 10^−2^	1.24 × 10^−1^
		*SD* [J/mm^2^]	4.76 × 10^−2^	7.45 × 10^−2^	2.79 × 10^−2^	5.98 × 10^−2^	4.61 × 10^−2^	2.96 × 10^−2^	4.46 × 10^−2^
	Score	*Avg. S* *	−0.22	1.67	−1.40	0.72	0.13	−0.81	−0.08
		*SD. S* **	0.03	1.68	−1.18	0.78	−0.06	−1.08	−0.16
		*LDS* ***	−12.33	2.94	−13.71	−1.42	9.66	11.38	3.48
ROI 12	SED	*Avg.* [J/mm^2^]	5.54 × 10^−2^	2.10 × 10^−1^	1.69 × 10^−1^	1.33 × 10^−1^	1.89 × 10^−1^	1.12 × 10^−1^	1.09 × 10^−1^
		*SD* [J/mm^2^]	3.66 × 10^−2^	7.38 × 10^−2^	1.13 × 10^−1^	4.48 × 10^−2^	8.21 × 10^−2^	4.90 × 10^−2^	3.44 × 10^−2^
	Score	*Avg. S* *	−1.58	1.32	0.55	−0.12	0.93	−0.52	−0.57
		*SD. S* **	−0.88	0.41	1.77	−0.59	0.70	−0.45	−0.95
		*LDS* ***	−37.51	47.17	−58.44	22.74	13.04	−4.44	17.43

* Average score; ** Standard deviation score; *** Load dependency score.

## Data Availability

No new data were created or analyzed in this study. Data sharing is not applicable to this article.

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
