# Peer review of "Quantitative Load Dependency Analysis of Local Trabecular Bone Microstructure to Understand the Spatial Characteristics in the Synthetic Proximal Femur"

_biology, 2023, doi:10.3390/biology12020170_

Round 1
Reviewer 1 Report
Summary: The objective of this study is to undertake a load dependency analysis of trabecular bone structure in the proximal femur. The authors used 12 ROIs placed within a 2D synthesized FE model of a proximal femur with trabecular structure model to be human-like.
1) There are too many abbreviations, it impacts the readability of the paper
2) The introduction oversimplifies and does not provide enough explanation about Wolff's Law. There is an extensive literature about Bone Functional Adaptation (see work by Timothy M. Ryan, Christopher Ruff, Jay Stock, Colin Shaw, Tracy Kivell, Jaap Saers, and Maureen Devlin for example). Bone is a very complex system, the structure and phenotypes is impacted by a lot more than loading. It is fine if that is the focus of the paper but it is important to provide a more fully fleshed out background about bone and what we know about understanding the relationship between form and function. Additionally, although Wolff formalized many of the ideas we think about related to Wolff's Law, he was not the first to make such observations (see Ward 1838 (Outlines of Human Osteology), von Meyer 1867 (Die architecktur der spongiosa) , Cowin 2001 (Bone Biomechanics Handbook)). You should also consider at the work done by H.M. Frost regarding the Mechanostat Theory of bone remodeling. In short, the current background does not provide adequate perspective on the importance of your research and is light on citations. Which makes it more difficult for the readers to understand why another femur FE paper is important.
3) Many of the studies cited are quite old, I'm sure there is also more recent literature that should be cited.
4) You should cite something for your discussion the inverse bone remodeling approach.
5) There are quite a few grammatical/spelling error. Often an extra or missing word. Or misplacement of words like "the" and "a". Careful editing to catch those errors should be undertaken.
6) Figure 3 seems to be missing in the manuscript
7) The whole manuscript seems quite light on citations.
8) The tables with the load dependency scores are a bit overwhelming, the use of scientific notation also makes it a bit more difficult to compare the scores. Is there a way this can be graphed so it is also more visual? I think that would help a lot with visualizing these results.
9) The discussion does a nice job of laying out why the results are important, I think this could be set up better in the introduction.
10) I think the inclusion of a limitations section was a really good choice. However, in line 422 you discuss "the precious study" which I don't think is what you meant. Maybe previous study? Probably better just to use the authors last names. Additionally, the last limitation you mention isn't explained at all.
11) The final sentence of the manuscript is quite vague. How will this work specifically help diagnose osteoporosis? How will your work help with fractures and other bone problems.
Reviewer 2 Report
The study was aimed to quantify the load dependency of the trabecular structure in different position inside the proximal femur. A 2D model for the proximal femur was generated using a Finite Element approach based on Topology Optimisation considering a weighted average of three different loading scenarios (i.e. 60% single leg stance, 20% abduction and 20% adduction). On the resulting structure (defined with the resolution of 50 microns) a FE analysis was run for 7 different loading scenarios; for each model (i.e. loading scenario) an indicator for the normalised Strain Energy Density was calculated for 12 different regions of interests positioned in anatomically relevant places. This indicator was able to quantify how dependent each ROI is on a considered loading scenario; conversely, this permits to determine what loading condition is the most effective in triggering remodelling on a specific ROI.
The manuscript is well structured, readable and well documented. Nevertheless, a few relevant points appear to critically necessitate a further elaboration from the Authors:
1) A synthetic 2D model generated by TO is used to study the SED in each region. Authors claim that this 2d approach has been already proven to bring to realistic trabecular architectures (“It was reported that the femur model exhibits the trabecular characteristic patterns”, rows137-138). Nevertheless, as far as the Reviewer know, a proper validation based on quantitative comparison between structures generated via TO and real structures identified with medical images is still missing. The aim of the study is to understand how dependent the trabecular structure is on the loading condition: a real trabecular structure should then be adopted. It is then unclear why Authors did not select for their analysis a 2D model derived from micro CT images of a real femur, and adopted instead a synthetic one bas on TO. In the Reviewer’s view, the quantitative comparison Authors proposed is based on a robust approach indicator; nevertheless - as the study is suggested now – the analysis is proving only how load dependent are different regions on a synthetic model generated via TO; the study does not prove how sensitive different regions are on a real femur to the loading conditions. Furthermore, the loading conditions used to generate the structure are also used to quantify the SED dependency; but the SED is also used to drive the TO process, bringing then to a potential bias. Coupling the present study with a similar analysis done on a “real” structure would further corroborate the strength of a TO approach, providing further confirmation of the load driven optimisation process that produced the articulated trabecular architecture. This comment approach a fundamental epistemological aspect that needs to be discussed and properly addressed.
2) Some of the parameters adopted in the study would need further discussion:
a. what is the size of the ROIs adopted for studying the SED values? What is/are the reasons for this choice?
b. What is the mesh sensitivity (i.e. how much the findings of the study depend on the model resolution?);
c. How and why the value of w in formula 5 was assumed to be 0.51? Is this value influencing massively the outcomes of the analysis?
Further comments
- Can the Authors provide some references for justify the values adopted for the elastic modulus E0 and the Poisson’s ratio of the bone tissue?
- Figure 3 appears missing, being only the caption being reported.
- Row 333: in view of the main comment, using the word “demonstrated” appears questionable. In the current form, an expression like “suggested” or “corroborated the hypothesis” appear a more appropriate option.
- The same consideration applies to row 350/351: in the Reviewer’s opinion, the current study does not indicate the validity of the adopted metric since a real validation does not offer a direct relevance for actual femoral structures.
- Especially in the perspective of a clinical application (i.e. likely dealing with pathological scenarios) mentioned at row 404, a 2D model is likely to capture the effect of mechanical/morphological animalities; the expression “it is clear that analysis using the 2D model is usefully meaningful” maybe appears a bit too strong (row 425).
Minor comments
- Authors might consider whether “loading conditions” would be more appropriated than “load conditions”.
- Table 2: how scores for the global model were calculated? Is the computation involving all the volume of the model or is that a simple summation of the values associated to the ROIs? Authors should consider to add a note to avoid possible confusion on this point.
